# Expression and role of lumican in acute aortic dissection: A human and mouse study

**Shao-Wei Chen[1,2], Shing-Hsien Chou[3], Ying-Chang Tung[3], Fu-Chih Hsiao[3], Chien-Te Ho[3], Yi-Hsin Chan[3], Victor Chien-Chia Wu[3], An-Hsun Chou[4], Ming-En Hsu[1,4], Pyng-Jing Lin[1], Winston W. Y. Kao[5], Pao-Hsien Chu[3]***

**1** Division of Thoracic and Cardiovascular Surgery, Department of Surgery, Chang Gung Memorial Hospital, Linkou Medical Center, Chang Gung University, Taoyuan City, Taiwan, **2** Center for Big Data Analytics and Statistics, Chang Gung Memorial Hospital, Linkou Medical Center, Taoyuan City, Taiwan, **3** Department of Cardiology, Chang Gung Memorial Hospital, Linkou Medical Center, Taoyuan City, Taiwan, **4** Department of Anesthesiology, Chang Gung Memorial Hospital, Linkou Medical Center, Chang Gung University, Taoyuan City, Taiwan, **5** Crawley Vision Research Center, Department of Ophthalmology, College of Medicine, University of Cincinnati, Cincinnati, Ohio, United States of America

* taipei.chu@gmail.com

**Data Availability Statement:** All relevant data are within the paper and its S1 Fig, S1 Raw images.

**Funding:** This work was supported by a grant from Chang Gung Memorial Hospital, Taiwan

## Abstract

### Introduction

Aortic dissection (AD) is a life-threatening emergency, and lumican (LUM) is a potential Bio-marker for AD diagnosis. We investigated LUM expression patterns in patients with AD and explored the molecular functions of Lum in AD mice model.

### Methods

LUM expression patterns were analyzed using aortic tissues of AD patients, and serum soluble LUM (s-LUM) levels were compared between patients with acute AD (AAD) and chronic AD (CAD). *Lum*-knockout (*Lum*$^{-/-}$) mice were challenged with β-aminopropionitrile (BAPN) and angiotensin II (Ang II) to induce AD. The survival rate, AD incidence, and aortic aneurysm (AA) in these mice were compared with those in BAPN–Ang II–challenged wild-type (*WT*) mice. Tgf-β/Smad2, Mmps, Lum, and Nox expression patterns were examined.

### Results

LUM expression was detected in the intima and media of the ascending aorta in patients with AAD. Serum s-LUM levels were significantly higher in patients with AAD than CAD. Furthermore, AD-associated mortality and thoracic aortic rupture incidence were significantly higher in the *Lum*$^{-/-}$ AD mice than in the *WT* AD mice. However, no significant pathologic changes in AA were observed in the *Lum*$^{-/-}$ AD mice compared with the *WT* AD mice. The BAPN–Ang II–challenged *WT* and *Lum*$^{-/-}$ AD mice had higher Tgf-β, p-Smad2, Mmp2, Mmp9, and Nox4 levels than those of non-AD mice. We also found that Lum expression was significantly higher in the BAPN-Ang II–challenged *WT* in comparison to the unchallenged *WT* mice.

CMRPG3K1951, CMRPG3J0661, BMRPD95
(SWC), CMRPG5K0011 (CTH), CMRPG3K0931
(SHC). This work was also supported by the
Ministry of Science and Technology grant Most
MOST-108-2314-B-182A-141 (SWC), MOST-106-
2314-B-182A-115-MY3 (PHC).

**Competing interests:** The authors have declared
that no competing interests exist.

## Conclusion

LUM expression was altered in patients with AD display increased s-LUM in blood, and
$Lum^{-/-}$ mice exhibited augmented AD pathogenesis. These findings support the notion that
LUM is a biomarker signifying the pathogenesis of injured aorta seen in AAD. The presence
of LUM is essential for maintenance of connective tissue integrity. Future studies should elu-
cidate the mechanisms underlying LUM association in aortic changes.

## Introduction

Aortic dissection (AD) is a life-threatening vascular emergency, with a sudden death rate of up
to 50% in type A AAD [1, 2], requiring immediate treatment for preventing aortic rupture.
Despite advancements in diagnostic, therapeutic, and surgical strategies, a study based on
International Registry of Acute Aortic Dissection, containing data of 4428 patients enrolled
over 1995–2013, revealed 22% and 14% mortality rates in patients with type A and B AAD,
respectively [2]. The key histopathologic feature of AD is medial degeneration, characterized
by smooth muscle cell (SMC) depletion and extracellular matrix (ECM) degradation [3]. Aor-
tic wall degeneration is possibly caused by abnormal gene expression and altered signaling
pathways, but the details of the underlying molecular mechanisms remain elusive [3]. There-
fore, gaining insight into the molecular and cellular mechanisms underlying this life-threaten-
ing disease is imperative for developing effective prevention and treatment regimens.

The small leucine-rich proteoglycans (SLRP) regulate cell migration and proliferation via
modulating collagenous matrix assembly [4]. SLRPs also bind collagen fibrils and regulate
fibril diameters and interfibrillar spacing of collagen in the ECM [4] implicating that they have
pivotal roles in cardiovascular diseases. For example, loss of biglycan (*BGN*), a member of
SLRP family, involves in aortic rupture, aortic aneurysm (AA), and AD observed in *BGN*-
knockout (KO) mice and patients with loss-of-function mutations in *BGN* [4–6]. Furthermore,
decorin, another SLRP, is associated with aortic SMC calcification [7].

LUM, also an SLRP family member, is particularly abundant in fibrotic tissues caused by
pathogenesis such as atherosclerosis and heart injury [4]. Interestingly, serum soluble LUM (s-
LUM) levels are elevated in patients with AAD [8–10] and thus it may serve as a biomarker for
AAD diagnosis. We previously demonstrated that LUM plays a protective role in cardiac fibro-
sis pathogenesis that characteristically associate with dysregulation of TGF-β/SMAD signaling
cascades and excessive MMP expression [11]. However, LUM expression patterns and under-
lying mechanistic molecular mechanism in the AD pathogenesis remain unclear.

Here, we investigated LUM expression in the aortic wall and serum from patients with
AAD and examined the underlying molecular mechanism by using an AD mouse models
including $Lum$-KO ($Lum^{-/-}$) mice and wild type (*WT*) mice manifesting pathogenesis due to
internal elastic lamina degeneration and hypertension via administration of β-aminopropioni-
trile (BAPN) and angiotensin II (Ang II), respectively.

## Methods

### Ethics statement

The handling and surgery protocols of experimental animals were approved by the Institu-
tional Animal Care and Use Committee of Chang Gung Memorial Hospital (Permit Number:
2019071601). Informed consent was obtained from all subjects in accordance with our

institution's guidelines and regulations. All surgical procedures of animals were performed under anesthesia with ketamine and xylazine, and all efforts were made to minimize animal suffering. Housing and maintenance were provided by Chang Gung Memorial Hospital, Taiwan; animals had ad libitum access to chow (Picolab 20, PMI Nutrition International) and water in accordance with the Lab guidelines and regulations of Chang Gung Memorial Hospital.

Use of human aortic tissue in this study was approved (201701920A3) by the Institutional Review Board of Chang Gung Memorial Hospital, Taiwan. We collected $5 \times 5$-cm$^2$-sized samples of the ascending aortas of patients with AAD. After excision, aortic tissue was flash-frozen in liquid $N_2$. Blood samples were collected from each patient who received surgery for AAD or CAD through indwelling arterial catheters on admission to the intensive care unit. Blood samples (10 mL) were collected in a syringe containing dry lithium heparin and centrifuged at $1000 \times g$ for 10 min. Plasma was immediately aliquoted, frozen on dry ice, and stored at $-80°C$ until analysis.

## Generation and maintenance of *Lum*$^{-/-}$ mice

*Lum*$^{-/-}$ mice were provided by Professors Kao and Chu [11, 12]. All animals used in this study (mice rendered homozygous null for *Lum* and their *WT* littermates) had been crossbred with C57BL/6 mice for >10 generations and genotyped through polymerase chain reaction. Western blot analysis was employed to demonstrate the lack of *Lum* expression in the protein extracted from the aorta and left ventricle of *Lum*$^{-/-}$ mice in our previous study [11].

## Mice AD model

C57BL/6 *Lum*$^{-/-}$ and *WT* mice were housed in standard cages with woodchip bedding and a paper roll for enrichment under constant ambient temperature (21–22°C) and humidity (40%–50%) with a 12/12 h light/dark cycle. The animal health and behavior were monitor twice a day. All animals had free access to tap water and the assigned diet. Special training in animal care and handling provided by our institution for all research staffs. For the induction of AD or aneurysms, 3-week-old male mice were fed on a regular diet and administered BAPN (Sigma-Aldrich, St. Louis, MO, USA) dissolved in drinking water (at 1 g/kg) per day for 4 weeks. At 7 weeks of age, *Lum*$^{-/-}$ and *WT* C57BL/6 mice in the study group received microosmotic pump implants (ALZET Osmotic Pump, Cupertino, CA, USA) filled with 1 μg/kg/min Ang II (Sigma-Aldrich), whereas those in the control group received osmotic minipumps filled with normal saline. Humane endpoints were set before the experiment, including reduced appetite (loss of appetite for 24 hours or reduced food intake >50% for 3 days), weakness (unable to stand for 24 hours), dying (depressive looking with decreasing body temperature), and organ failure (such as difficulty in breathing, cyanosis, severe vomiting, and self-injurious behavior). Once the preceding symptoms were observed, the mice were euthanized with $CO_2$ immediately. The mice were euthanized 28 days after implantation with $CO_2$ inhalation. The aortas of these mice visualized before minipump implantation and before sacrifice through 7T magnetic resonance imaging (MRI). The experiment was performed from 2019 January to 2020 July.

## Tissue preparation and evaluation of AA, AD, and aortic rupture in mice

Animals were anesthetized and thoracotomy was performed to insert a perfusion catheter into the left ventricle, after which blood samples were collected. The mice were subsequently exsanguinated through a right atrial incision and perfused with cold sodium pentobarbital or KCl. Aortic tissue from the ascending aorta to the iliac bifurcation was resected, frozen in liquid $N_2$,

and stored until further analysis. We evaluated the ascending arch, descending thoracic, and abdominal aortic segments in each extracted aorta, and the diameter of each aortic segment was measured. AA was defined as an aortic diameter > 1.5 times the mean aortic diameter of the segment in unchallenged, vehicle-treated mice, whereas AD was defined as the presence of layer separation within the aortic media or adventitial boundary or the presence of hematoma within the aortic wall detected on gross or histological examination. Death due to aortic rupture was recorded.

## Morphometry, immunohistochemical staining, and microscopy of the aorta

Human and mice aortas were fixed with 10% buffered formalin and embedded in paraffin. Sections of blood vessels collected at baseline were stained using hematoxylin–eosin. We used Verhoeff–Van Gieson (VVG) stain for elastic fibers and Masson's trichrome stain for collagen and elastic fibers. LUM was detected in human using lumican antibody (R&D, AF2846) according to the manufacturer's recommendations.

## Western blot analysis

Protein was quantified after extraction from RIPA lysis buffer (25 mM Tris, 150 mM Sodium Chloride, 1% NP-40, 1% Sodium Deoxycholate, 0.1% SDS, pH 7.6). Next, 10 μg of the protein homogenate was mixed with an adequate volume of Laemmli sample buffer and subjected to sodium dodecyl sulfate (SDS) polyacrylamide gel electrophoresis (10–12% gel). Gels were blotted onto polyvinylidene difluoride (PVDF) membranes (Merck Millipore Ltd., Tullagreen, Carrigtwohill,Co. Cork IRL). Membranes were washed thrice in Tris-buffered saline containing 2% (vol/vol) Tween 20 (TBST) for 10 min at room temperature (22 ˚C) and incubated in blocking solution (5% [wt/vol] nonfat dried skimmed milk powder in TBST) for 1 h at room temperature. Next, the blots were incubated with a primary antibody in the blocking solution overnight at 4 ˚C with slow rocking. The blots were washed thrice for 10 min in TBST and incubated with a horseradish peroxidase-conjugated secondary antibody in the blocking solution. Thereafter, the bands were identified through chemiluminescence. The membranes were stripped using Stripping Buffer (BIONOVAS biotechnology Co., Ltd., Toronto, Ontario), according to the manufacturer's instructions and reblotted for Actin. Band intensities were quantified using Image J. Data were expressed as arbitrary units relative to Actin levels. Lum, Tgf-β, phospho-Smad2 (pSmad2), Mmp2, Mmp9, Nox2, and Mmp14 levels were determined using Western blot analysis.

## Enzyme-linked immunosorbent assay

Human serum s-LUM levels were determined using an enzyme-linked immunosorbent assay (ELISA; Cusabio Biotech, Wuhan, China) according to manufacturer's recommendations. Signals from a 96-well plate were read using an ELISA reader.

## Statistical analysis

The results were expressed as mean ± standard error. Student $t$ test and Mann–Whitney $U$ test were used to determine the significance of variable differences between the groups. The $\chi^2$ test or Fisher exact test was used to compare categorical data. Kaplan–Meier survival curves were plotted to analyze the mouse survival rates, and the differences were analyzed using the log-rank test. Significance was established at $P < 0.05$. All statistical analyses were performed on SPSS for Windows (version 16.0; SPSS, Chicago, IL, USA).

## Results

### LUM expression in patients with AAD

The ascending aortas of patients with AAD were subjected to immunohistochemical staining for determining LUM expression (Fig 1 and S1 Fig). LUM expression was particularly high in the intima and media. In addition, ELISA assay was performed to determine the amount of sLUM in serum of AAD and CAD patients. Fig 2 shows average serum s-LUM levels in patients with AAD (n = 14) and those with CAD and requiring surgery (n = 3) were 3.44 and 1.02 ng/mL, respectively. Therefore, patients with AAD had 3.37-fold higher s-LUM levels than did those with CAD ($p$ = 0.013; Fig 2).

### Survival rates and aortic rupture rates in $Lum^{-/-}$ and WT mice challenged with BAPN-Ang II

To determine the role of Lum in AD, we subjected $Lum^{-/-}$ and $WT$ mice to a BAPN (1 g/kg/day of 3-aminopropionitrile fumarate salt metabolite) diet for 4 weeks and then implanted a minipump for Ang II (1 µg/kg/min) challenge for another 4 weeks. Without the BAPN–Ang II challenge, no mice died in either group. After the BAPN–Ang II challenge, 26.67% (4/15) of the $WT$ mice and 81.82% (18/22) of the $Lum^{-/-}$ mice died (Fig 3A). Most cases of aortic dissection and sudden death occurred within 1 week after Ang II challenge. These mice died suddenly due to aortic dissection or aortic rupture, without the symptoms of predetermined humane endpoints; hence, euthanasia could not be performed in time. The $Lum^{-/-}$ mice exhibited a significantly higher mortality risk due to AD and aortic rupture than did the $WT$ mice ($p$ = 0.003; Fig 3A).

Moreover, 13.64% (3/22) and 6.67% (1/15) of the $Lum^{-/-}$ and $WT$ mice, respectively, challenged with BAPN–Ang II had abdominal aortic rupture. In contrast, thoracic aortic ruptures were observed in 77.27% (17/22) and none (0/15) of the $Lum^{-/-}$ and $WT$ mice challenged with BAPN–Ang II, respectively (p = 0.006; Fig 3B). A histologic analysis of the AD and aortic rupture in the $Lum^{-/-}$ mice challenged with BAPN–Ang II with hematoxylin–eosin staining, silver stain, and Verhoeff–van Gieson elastic and reticular fiber staining revealed severe destruction and elastic fiber fragmentation in the aortas (Fig 4).

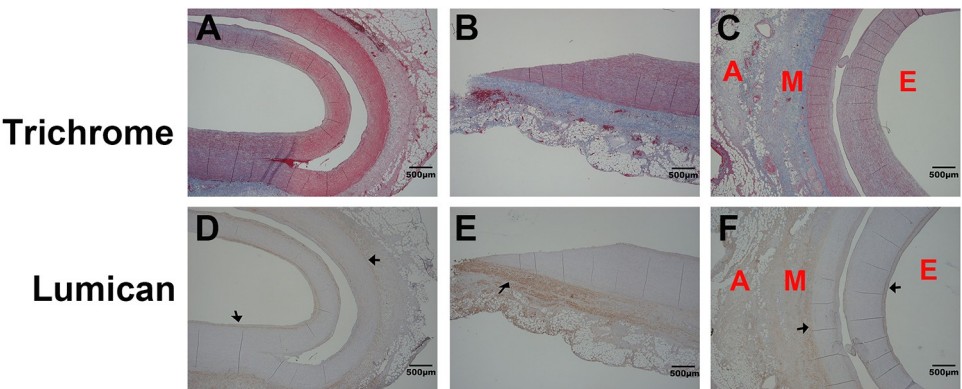

**Fig 1. Lum expression patterns in the ascending aorta of patients with acute aortic dissection (AAD).** (A–C) Representative images of human ascending aorta stained with Masson's trichrome, revealing patterns of collagen and elastic fibers. A, M, and E represent the adventitia, media, and endothelium layers, respectively. (D–F) Corresponding sections adjacent to the A–C sections and immunohistochemically stained with the human LUM antibody (R&D, AF2846). Black arrows indicate presence of LUM in the aortic intima and media. The size bar is 500 µm.

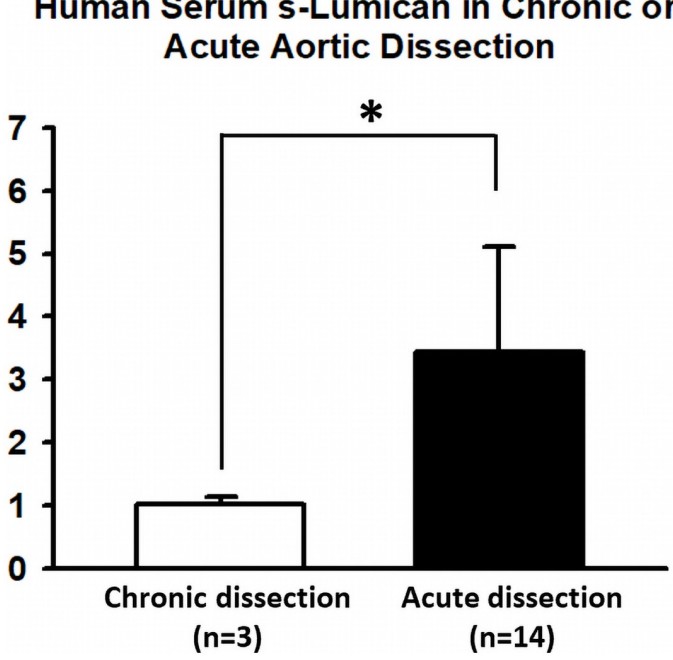

**Fig 2. Serum s-LUM levels in patients with acute aortic dissection (AAD) and chronic aortic dissection (CAD).** Human serum s-LUM levels were determined using an enzyme-linked immunosorbent assay. The average serum s-LUM levels were significantly (3.37-fold) higher in patients with AAD than in those with CAD (3.44 vs. 1.02 ng/mL, $p = 0.013$).

## Evaluation and comparison of size and pathological features of the aorta between $Lum^{-/-}$ and WT mice

The variegation of size and lesion along the damaged aorta in both BAPN–Ang II–challenged $Lum^{-/-}$ and $WT$ mice at various stages before and four weeks after Ang II minipump implantation, the pathogenesis of the disease were periodically evaluated by noninvasive 7T-MRI. As shown in Fig 5, despite the AD and aortic rupture rates being higher in the BAPN–Ang II–challenged $Lum^{-/-}$ mice than in the BAPN–Ang II–challenged $WT$ mice, the imaging findings did not vary significantly between the groups. As presented in Fig 3, most cases of aortic dissection and sudden death occurred within 1 week after Ang II challenge. The median time window from the last MRI scan to sudden death was 2–3 days. The BAPN–Ang II challenge increased AD-associated mortality and thoracic aortic ruptures in $Lum^{-/-}$ mice compared with $WT$ mice, but the aneurysm change of the two groups did not vary significantly. The observation implicates that the aortic pathogenesis in thoracic aorta caused by the lack of Lum may account for AD-associated mortality and aortic rupture without obviously association of AA.

## Comparison of Tgf-β/pSmad2, Mmp9, Mmp2, Nox4, and Lum levels between experimental and control mice

Aortic tissue from the ascending aorta to the iliac bifurcation of the mice was resected and used for analysis. The BAPN–Ang II challenge upregulated Tgf-β and pSmad2 expression in both the $Lum^{-/-}$ and $WT$ mice (Fig 6A). In the control group, the $Lum^{-/-}$ mice had significantly higher pSmad2 levels than that of the $WT$ mice (Fig 6B). Expression of Mmp2, Mmp9,

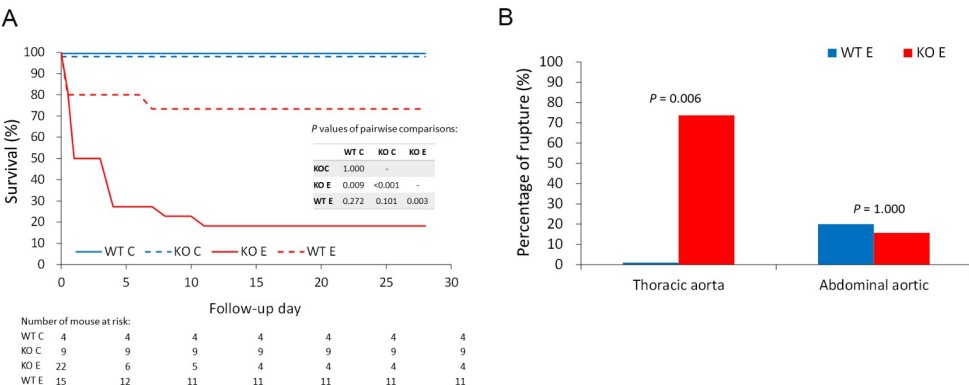

**Fig 3. Comparison of survival and aortic rupture rates between *Lum*$^{-/-}$ and *WT* mice with or without the BAPN–Ang II challenge.** (A) Kaplan–Meier survival analysis was performed to compare survival between *WT* mice without the BAPN–Ang II challenge (WT C), *Lum*$^{-/-}$ mice without the BAPN–Ang II challenge (KO C), *WT* mice with the BAPN–Ang II challenge (WT E), and *Lum*$^{-/-}$ mice with the BAPN–Ang II challenge (KO E). Numbers of days and mice that were followed are listed below the survival curve. Results of the pairwise comparison using log-rank test are also presented. (B) Aortic rupture rates in either the thoracic aorta or abdominal aorta were compared between *WT* mice challenged with BAPN–Ang II (WT E) and *Lum*$^{-/-}$ mice challenged with BAPN–Ang II (KO E).

and Nox4 levels were significantly higher in the BAPN–Ang II–challenged AD mice than in the control mice. However, Mmp2, Mmp9, Mmp14, and Nox4 levels were not significantly different between the naïve unchallenged *Lum*$^{-/-}$ and *WT* mice (Fig 6C–6F, respectively).

We also assessed Lum protein expression patterns in *WT* mice with or without BAPN-Ang II challenges (Fig 7A) and found that Lum expression was significantly up regulated by 2.1-fold (p < 0.05) in the BAPN-Ang II–challenged *WT* mice in comparison to the unchallenged *WT* mice (Fig 7B).

## Discussion

In this study, we demonstrated that LUM was expressed in intimal and medial layers of the ascending aorta of patients with AAD. Patients with AAD had higher serum s-LUM levels than that of patients with CAD. The AD mouse model further demonstrated that lack of Lum contributed to increased risks of AD-related mortality and thoracic aortic rupture and altered Tgf-β/Smad signaling and Mmps expression, but the aneurysm change of the two groups did not vary significantly. Therefore, our observations are consistent to the notion that aortic pathogenesis due to the lack of Lum might only occur in thoracic aorta and might be directly involved in the events related to AD-associated mortality and aortic rupture without obviously influencing AA morphology. We also found that Lum expression was significantly higher in the BAPN-Ang II–challenged *WT* than naive *WT* mice. Taken together, our findings suggest that Lumican is important to maintain the aortic structure to prevent aortic dissection and may be crucial in alleviating AD pathogenesis in that *Lum*$^{-/-}$ mice display severe pathogenesis upon induction of AAD by BAPN-Ang II challenge, which weakens cross-link of collagen fibrils in ECM. We speculate that LUM increase to response the aortic injury as a healing compensation both in human and mice. Therefore, the Increase of s-LUM is an important clinical biomarker of AD. However, further studies for explaining the regulation mechanism of LUM and clarifying its role in the human aorta are warranted.

Results of current study indicate that LUM expression patterns are in the intima and media of the ascending aorta of patients with AAD. Gu et al. identified LUM as a potential serum marker by using quantitative proteomics [8]. They further observed that LUM is expressed in

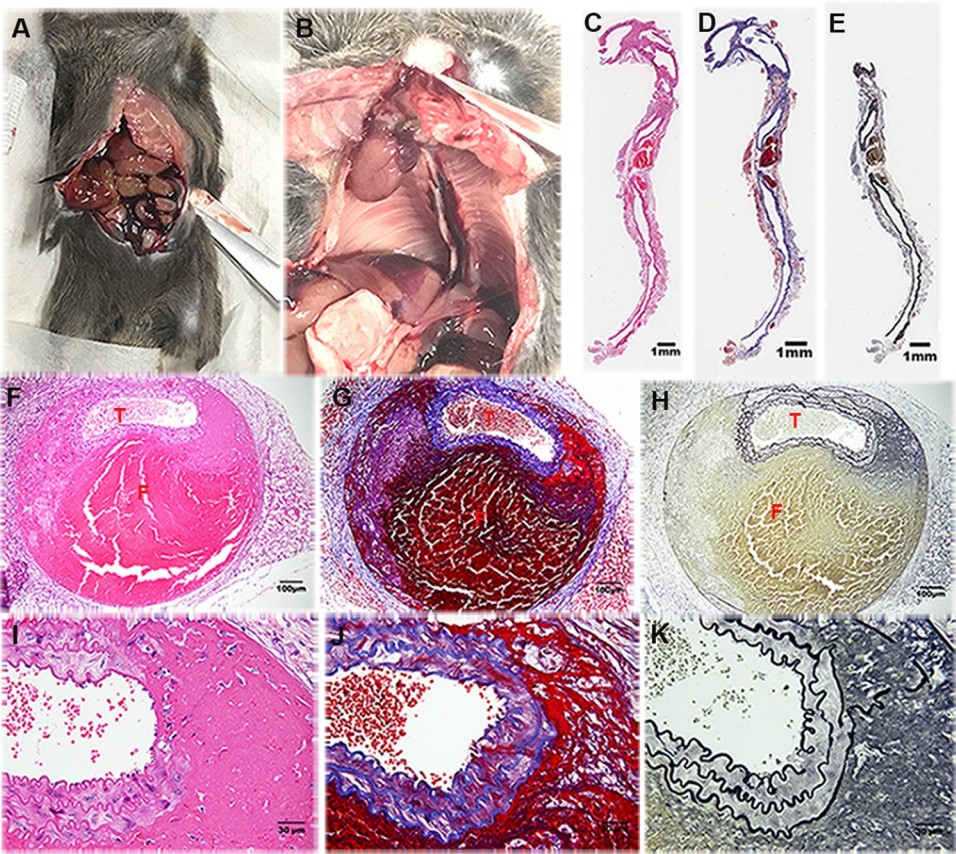

**Fig 4. Structures of aortic dissection (AD) in *Lum*⁻/⁻ mice challenged with BAPN–Ang II.** Representative photographs of (A) abdominal AD and aortic rupture and (B) thoracic AD and aortic rupture. (C–K) Representative microscopy images of sagittal sections of the ruptured aorta stained with hematoxylin–eosin, Masson's trichrome, and Verhoeff–Van Gieson stains at 5× magnification (C, D, and E, respectively. The size bar is 1mm), at 20× magnification (F, G, and H, respectively. The size bar is 100 μm), and at 400× magnification (I, J, and K, respectively. The size bar is 30 μm). (T: True lumen; F: False lumen).

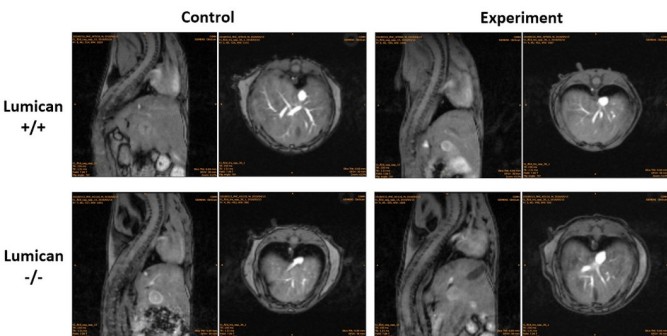

**Fig 5. MRI examination of the size and pathological position of the aorta in both BAPN–Ang II–challenged *Lum*⁻/⁻ and *WT* mice.** Representative images of noninvasive 7T MRI scanning of the aortas 4 weeks after micropump implant and before euthanasia in *Lum*⁻/⁻ and *WT* (*Lum*⁺/⁺) mice with (Experiment) or without the BAPN–Ang II challenge (Control).

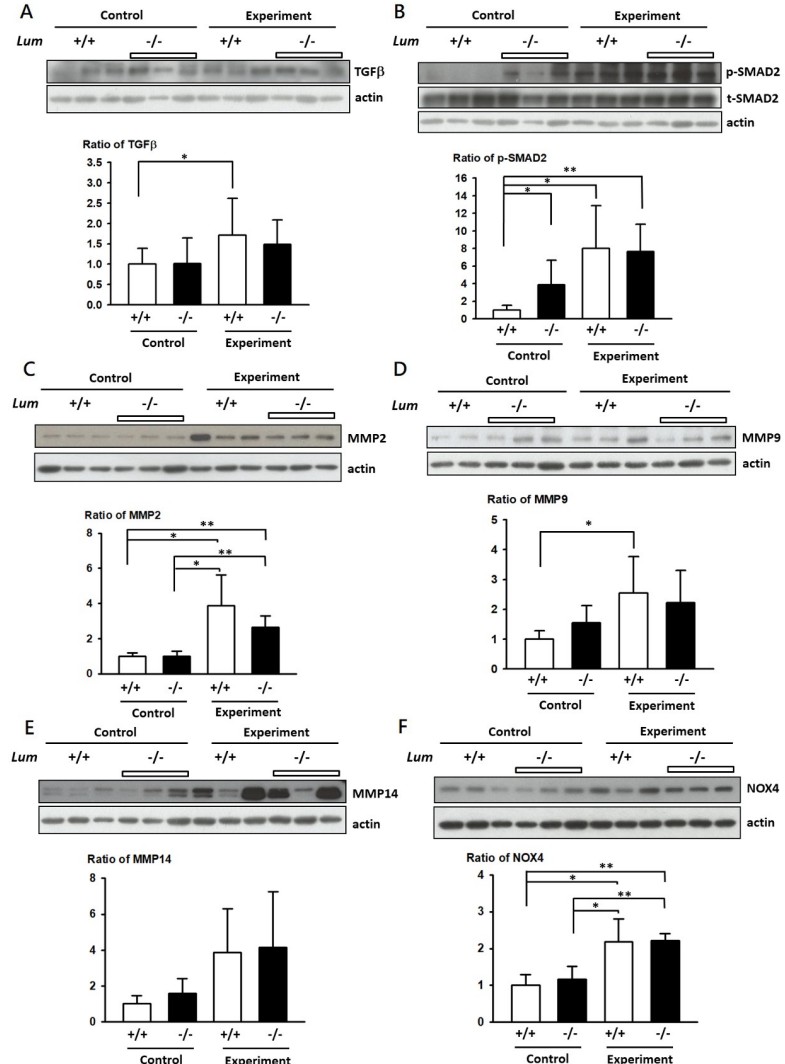

**Fig 6. Variation of TGFβ signaling and MMPs expression in *Lum*^−/−^ and *WT* mice by BAPN–Ang II challenge.**
Western blot analysis of TGF-β (A), pSMAD2/ tSMAD2 (B), MMP2 (C), MMP9 (D), MMP14 (E), and NOX4 (F)
levels in *Lum*^−/−^ and *WT* (*Lum*^+/+^) mice with (Experiment) or without the BAPN–Ang II challenge (Control).
Expression intensities were quantified as level of expression was calibrated with actin intensity units relative to actin
levels.

the aortic medial layer in both healthy donors and patients with AAD [9]. However, LUM
expression in the intimal layer was not reported. Another study using a mouse model to exam-
ine transcriptomic changes during aortic aneurysm development in Marfan syndrome identi-
fied differential expression patterns of LUM in SMC [13]. Other studies on LUM expression
have focused on atherosclerosis and arterial calcification. A study reported LUM expression in
the outer layer of SMCs and atheromatous plaques in patients with coronary atherosclerosis
[14]. Consistent with our findings, this study demonstrated that LUM synthesis occurs mainly
in coronary artery intima and media [14]. In the peripheral arteries, LUM expression was sig-
nificantly higher in the intima of atherosclerosis-prone internal carotid artery than in the
intima of the atherosclerosis-resistant internal thoracic artery, thereby indicating its role in
atherosclerosis pathogenesis [15]. Patients with chronic kidney disease had an increased risk

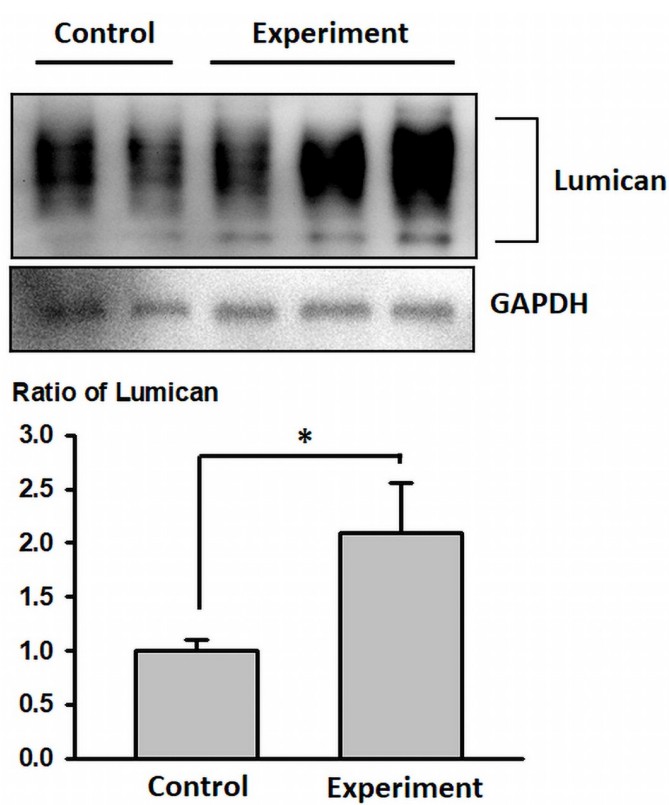

**Fig 7. Lumican expression patterns in *WT* mice with or without the BAPN–Ang II challenge.** Western blot analysis of lumican expression in *WT* mice with the BAPN–Ang II challenge (Experiment) or without the BAPN–Ang II challenge (Control). Expression intensities were quantified as arbitrary units relative to the GAPDH levels.

of arterial calcification and exhibited concomitant increased LUM expression [16]. Because both atherosclerosis and arterial calcification are linked to AD [17, 18], LUM might participate in this pathogenesis through collagen fibrogenesis regulation and SMC migration and proliferation.

Several studies have advocated s-LUM as an AAD biomarker. Higher s-LUM serum levels were observed in patients with AAD than in healthy people or patients without AD [8–10]. In the present study, patients with AAD had significantly higher serum s-LUM levels than did those with CAD. Thus, our findings strengthened s-LUM use as an AAD biomarker by demonstrating a relationship of s-LUM levels with disease stage. However, whether the differences of serum s-LUM levels between patients with AAD and CAD result from differences in LUM expression in aorta or differences time course in its release into the serum in AD remains unclear. It will also be interesting to know if s-lumican expressed and/or release more in aortic rupture. Future studies should elucidate the correlation between tissue expression, serum level, and AD severity and clinical outcomes to facilitate the use of s-LUM as an AD biomarker.

We also observed differences in thoracic and abdominal aortic ruptures between $Lum^{-/-}$ and *WT* mice subjected to BAPN-Ang II challenge, potentially due to structural differences between the thoracic and abdominal aneurysms [19]. Although the BAPN–Ang II challenge increased AD-associated mortality and thoracic aortic ruptures in $Lum^{-/-}$ mice compared with *WT* mice, aneurysm change between the two groups did not vary significantly. We speculate that aortic pathogenesis due to the lack of LUM might only occur in thoracic AA or might

be directly involved in the events related to AD-associated mortality and aortic rupture without obviously influencing AA morphology. Moreover, AD-associated morality and thoracic aortic rupture may occur in regions of aorta not affected by AA. This phenomenon is similar to the type of aortic disease prevalent in the Asian population, different from that found in the Western population, where patients generally develop AA before AD. The relevant genetic and ethnic differences imply that AD-related mortality, aortic rupture, and AA pathogenesis might differ under different genetic regulations and that LUM is potentially involved only in some specific pathways. Further research exploring the causes of AD-related mortality is warranted for gaining insight into the role of LUM in AD pathogenesis.

In our study, we observed elevated p-SMAD in $Lum^{-/-}$ mice compared with $WT$ mice. We however did not see further p-SMAD or TGF-β signal enhanced in $Lum^{-/-}$ mice upon BAPN–Ang II-challenging. Both BAPN–Ang II-challenged $Lum^{-/-}$ and $WT$ AD groups exhibited elevated TGF-β and SMAD levels. TGF-β/SMAD signal transduction is a core pathophysiological pathway, responsible for inducing ECM deposition and repressing ECM degradation; thus, alterations in TGF-β/SMAD signaling might be crucial in AD pathogenesis [20]. However, review of multiple publications also revealed that both overexpression and downregulation of TGF-β/SMAD signaling can promote AA formation due to activation of non-canonical TGF-β signaling upon dysregulation of TGF-β/SMAD signaling [21]. Therefore, it is possible under BAPN–Ang II-challenging, effects of $Lum$ KO cannot be revealed due to activation of bypass non-canonical signaling. It is also possible that, LUM and TGF-β/SMAD signaling are involved in the same pathway and TGF-β/SMAD signaling is downstream of LUM. Future experiments that further manipulate these gene can help answering this question.

We demonstrated significantly higher MMP levels in the BAPN–Ang II-challenged $Lum^{-/-}$ and $WT$ AD mice than in the control group. The finding is consistent with studies indicating the release of MMPs, particularly MMP9, in response to Ang II and their role in AD induction through ECM degradation [22–24]. Further understanding of the genetic and molecular mechanisms of AD could improve future disease prevention and treatment strategies. Investigating the role of LUM is a critical field of research in cancer molecular biology because it affects MMP regulation and has been demonstrated to inhibit MMP9 and MMP14 expression [25]. Moreover, LUM plays protection roles in cancer through inhibiting MT1-MMP in melanoma cells [26]. Therefore, considering the relationship of AD with MMPs, particularly MMP2, MMP9, and MMP14, LUM can be an effective regulator on catabolism of ECM and used in the development of new treatment strategies [27]. Nevertheless, further are needed to explore the role of LUM in TGF-β/SMAD signaling and TGF-β-induced MMP inhibition.

NOX increases endothelial superoxide generation and elastic lamina fragmentation in response to Ang II, thus increasing susceptibility to AD [28]. BGN has also been shown to play a role in NOX2 synthesis and activation [29]. We demonstrated a significantly higher NOX level in the Ang II-challenged $Lum^{-/-}$ and $WT$ AD mice than in the control group. However, we have not evaluated the levels of superoxide and other reactive oxygen species. Thus, the role of LUM in NOX regulation and AD induction remains unclear. Additional studies clarifying the underlying mechanisms are required.

Our study has some limitations. First, although a high AD-associated mortality rate was observed in $Lum^{-/-}$ mice, we cannot evaluate AD and associated mortality in human, because LUM mutation has not been associated with human aortic diseases. Future research may include sequencing the $LUM\ allele$ from AAD patient's tissues or blood. Second, our findings do not convey the essential role of LUM in the human aorta and lack the documentation of the mechanism through which it reaches the bloodstream in patients with AD, though our observation of $Lum^{-/-}$ mice are consistent that Lum may have a pivotal role in ameliorating the symptoms of AAD. Third, no significant difference was observed in TGF-β, SMAD, MMP,

and NOX levels between $Lum^{-/-}$ and $WT$ mice, how LUM mediates TGF-β, SMAD, MMP, and NOX expression remains unclear. Future studies elucidating the mechanisms of LUM are warranted to enhance our understanding of its potential role in AD.

## Conclusion

LUM expression was altered in patients with AD display increased s-LUM in blood, and $Lum^{-/-}$ mice exhibited augmented AD pathogenesis. These findings support the notion that LUM is a biomarker signifying the pathogenesis of injured aorta seen in AAD. The presence of LUM is essential for maintenance of connective tissue integrity. Future studies should elucidate the mechanisms underlying LUM association in aortic changes.

## Supporting information

**S1 Fig. Negative control of LUM immunohistochemical stain of the aorta in patients with AAD.**
(TIF)

**S1 Raw images.**
(PDF)

## Author Contributions

**Formal analysis:** Shao-Wei Chen.

**Methodology:** Shao-Wei Chen, Shing-Hsien Chou, Ying-Chang Tung.

**Visualization:** Shing-Hsien Chou, Fu-Chih Hsiao, Chien-Te Ho, Yi-Hsin Chan, Victor Chien-Chia Wu, An-Hsun Chou, Ming-En Hsu, Pyng-Jing Lin, Winston W. Y. Kao.

**Writing – original draft:** Shao-Wei Chen.

**Writing – review & editing:** Pao-Hsien Chu.

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
