## [Decision Letter · Decision Letter 0]

23 Mar 2021

PONE-D-21-03961

Expression and Role of Lumican in Acute Aortic Dissection: A Human and Mouse Study

PLOS ONE

Dear Dr. Chu,

Thank you for submitting your manuscript to PLOS ONE. Based on careful evaluation by two expert reviewers your manuscript is considered to have merit, but does not fully meet PLOS ONE’s publication criteria as it currently stands. Therefore, we invite you to submit a revised version of the manuscript that addresses the points raised during the review process.

Please also make sure that the manuscript follows the PLoS ONE policy regarding the availability of all data. This is very important.

A rebuttal letter that responds to each point raised by the academic editor and reviewer(s). *The revised text should be copied to this letter and its location (page and line numbers) indicated. *You should upload this letter as a separate file labeled 'Response to Reviewers'.A marked-up copy of your manuscript that highlights changes made to the original version. You should upload this as a separate file labeled 'Revised Manuscript with Track Changes'.An unmarked version of your revised paper without tracked changes. You should upload this as a separate file labeled 'Manuscript'.

We look forward to receiving your revised manuscript.

Kind regards,

Helena Kuivaniemi, MD, PhD

Academic Editor

PLOS ONE

Journal Requirements:

Additional Editor Comments (if provided):

Reviewers' comments:

Reviewer's Responses to Questions

**Comments to the Author**

1. Is the manuscript technically sound, and do the data support the conclusions?

Reviewer #1: Partly

Reviewer #2: Yes

2. Has the statistical analysis been performed appropriately and rigorously? 

Reviewer #1: I Don't Know

Reviewer #2: Yes

3. Have the authors made all data underlying the findings in their manuscript fully available?

Reviewer #1: Yes

Reviewer #2: Yes

4. Is the manuscript presented in an intelligible fashion and written in standard English?

Reviewer #1: Yes

Reviewer #2: Yes

5. Review Comments to the Author

Reviewer #1: The expression level of LUM and s-LUM is known to be significantly higher in aortic tissues and the serum of patients with AD (PMID, 22228989, 26998013, 32698686). Consistent with previous results, the author reported here that Lum expression level in serum and aneurysm tissues is higher in patients with AD, suggesting that Lum may be associated with the pathogenesis of AD. Although it is not explicitly stated, the authors explored the functional role of LUM in Lum-/- mice. The authors found that Lum−/− mice exhibited augmented severity of AD in response to BAPN and AngII treatment and accompanied by increased expression of pSMAD2. Overall, the MS is logical. However, the abstract is vague; the methods/results and figure legends are not detailed enough. There is no explicit conclusion stated in the abstract. There is only vague speculation. The analysis was not described in detail. The images did not show enough details of the vessel wall. Here are specific comments.

1. The title is misleading: the role or molecular function of Lumican was not investigated.

2. Abstract: please move “Lum-knockout (Lum−/−) mice were challenged with β-aminopropionitrile (BAPN) and angiotensin II (Ang II) to induce AD.” to line 47.

3. Fig1 and 3: lack scale bar. Please add the topic sentence to each figure legend. For IHC, please include negative controls using IgG and the secondary antibody alone. Please confirm that the LUM antibody (indicate source) is Lum specific by WB assay.

4. Methods: “electron microscopy” is a typo. Please correct the term in method and result sections.

5. Fig 4B. High magnification images of H&E, elastin and collagen staining should be included.

6. Line 211: Please describe the observation by the VVG and silver staining.

7. Fig 5. It is unclear about the periodic evaluation used for MRI. The negative result may be due to the missed window of aneurysm progression. How fast did the dissection progress? What was the time from the last scan to sudden death?

8. Figure 6. How long was the treatment in experimental group? Are mice in Fig 4 and 6 from the same experimental treatment group? Please show representative WB images. IHC assay should be informative.

9. Line 236 in Discussion: “To the best of our knowledge, this is the first study to demonstrate LUM expression patterns in patients with AAD“. Please discuss PMID 22228989, 26998013, 32698686.

10. Fig 6: Was the expression of Lum increased in the aorta wall and the serum from the experimental WT group? Was Lum expression level associated with the risk/severity of AD in the WT experimental group?

Reviewer #2: The manuscript entitled "Expression and Role of Lumican in Acute Aortic Dissection: A Human and Mouse Study" by Chen et al examined LUM expression 40 patterns in patients with AD and explored the molecular functions of LUM in a mouse AD model. The authors concluded that LUM expression is altered in patients with AAD, and Lum−/− mice exhibited augmented AD pathogenesis and LUM may be a novel target for AD therapy.

While the concept of the paper is of high importance towards the possibility of Lum as novel therapeutic target for AAA, There are major concerns with the manuscript:

1. While differences in Lum expression was observed between acute AD (AAD) and chronic AD (CAD), it is not clear how these tissue samples was characterized. Also, in Fig. 1, the staining is not clear and resolution is poor. Infact, in Fig. 1F, it seems that expression of Lum is increased in the adventitial layer.

2. Figs. 4C-F in the present form is not adding much to the data. The authors need to add high magnification images and interpretation of the results.

3. The authors reported higher TAD in the Lum KO mice, how does it correlate to the human AADs is not clear.

4. It seems that most of the mice (18/22) died of TAD. It is not clear how the remaining mice were used.

5. It is not clear which part of the tissue (aorta?) and how many were used for Fig 6A. It is also not clear whether the authors used abdominal or the whole aorta.

6. PLOS authors have the option to publish the peer review history of their article (what does this mean?). If published, this will include your full peer review and any attached files.

Reviewer #1: No

Reviewer #2: No

---

## [Author Response · Author response to Decision Letter 0]

25 May 2021

May 22, 2021

Editor-in-Chief

PLOS ONE

PONE-D-21-03961 Expression and Role of Lumican in Acute Aortic Dissection: A Human and Mouse Study

Dear Editor:

On behalf of the authors of this manuscript, we thank you, associate editor and 2 reviewers very much for your thorough review of this manuscript. Your suggestions were very thoughtful and instructive. The reviewers identified some serious and important concerns that we have not considered while making this study. We revised the manuscript according to the comments of the reviewers’ point-by-point as follows. We hope that the revised manuscript will retain your attention and you will judge the revised manuscript to be suitable for publication in “PLOS ONE”. Please forward the revised manuscript to the reviewers and reconsider the work for publication. 

Yours sincerely,

First authors: Shao-Wei Chen 

Corresponding author: Pao-Hsien Chu

Reviewer #1: The expression level of LUM and s-LUM is known to be significantly higher in aortic tissues and the serum of patients with AD (PMID, 22228989, 26998013, 32698686). Consistent with previous results, the author reported here that Lum expression level in serum and aneurysm tissues is higher in patients with AD, suggesting that Lum may be associated with the pathogenesis of AD. Although it is not explicitly stated, the authors explored the functional role of LUM in Lum-/- mice. The authors found that Lum−/− mice exhibited augmented severity of AD in response to BAPN and AngII treatment and accompanied by increased expression of pSMAD2. Overall, the MS is logical. However, the abstract is vague; the methods/results and figure legends are not detailed enough. There is no explicit conclusion stated in the abstract. There is only vague speculation. The analysis was not described in detail. The images did not show enough details of the vessel wall. Here are specific comments.

Reply: Thank you for your helpful comments. We have revised our abstract, the methods/results section, and figures in accordance with your suggestions. 

1. The title is misleading: the role or molecular function of Lumican was not investigated.

Reply: Thank you for your comment, we think our title “Expression and Role of Lumican in Acute Aortic Dissection: A Human and Mouse Study” is to describe our finding of lumican in aortic dissection both in human and mice experiment, not specific meaning the role of molecular function. However, if the reviewer insistent to delete the “role “, we will remove this word from the title. 

2. Abstract: please move “Lum-knockout (Lum−/−) mice were challenged with β-aminopropionitrile (BAPN) and angiotensin II (Ang II) to induce AD.” to line 47.

Reply: We have moved the sentence as suggested. 

3. Fig1 and 3: lack scale bar. Please add the topic sentence to each figure legend. For IHC, please include negative controls using IgG and the secondary antibody alone. Please confirm that the LUM antibody (indicate source) is Lum specific by WB assay.

Reply: We have revised our Figures 1 and 3 as indicated. We have added the scale bar and topic sentence in Figures 1 and 3. Furthermore, we have provided the data on the negative control of LUM immunohistochemical stain of the aorta in patients with acute aortic dissection (Supplementary Figure 1: Negative control of LUM immunohistochemical stain of the aorta in patients with AAD). We have specified that the LUM antibody for immunohistochemistry and WB in human aortic tissue was obtained from R&D (AF2846) and is specific for humans.

Page 8 line159: LUM was detected in human using lumican antibody (R&D, AF2846) according to the manufacturer’s recommendations.

4. Methods: “electron microscopy” is a typo. Please correct the term in method and result sections.

Reply: We apologize for the typo. We have deleted all the mentions of electron from our manuscript. 

5. Fig 4B. High magnification images of H&E, elastin and collagen staining should be included.

Reply: We have revised Figure 4 and provided high-magnification images. The baseline sections of blood vessels were stained using hematoxylin–eosin. We used Verhoeff–Van Gieson stain for elastic fibers and Masson’s trichrome stain for collagen and elastic fibers. 

6. Line 211: Please describe the observation by the VVG and silver staining.

Reply: We have added the descriptions of the observations obtained from the HE, VVG, and Masson’s trichrome staining in our results section.

Page 11 Line 219: A histologic analysis of the AD and aortic rupture in the Lum−/− mice challenged with BAPN–Ang II with hematoxylin–eosin staining, silver stain, and Verhoeff–van Gieson elastic and reticular fiber staining revealed severe destruction and elastic fiber fragmentation in the aortas (Fig. 4). 

7. Fig 5. It is unclear about the periodic evaluation used for MRI. The negative result may be due to the missed window of aneurysm progression. How fast did the dissection progress? What was the time from the last scan to sudden death?

Reply: The aortas of these mice were visualized at two time points through 7T magnetic resonance imaging (MRI): before minipump implantation (BAPN feeding for 4 weeks) and before killing (Ang II challenging for 4 weeks). 

As presented in Figure 3 (survival curve), most cases of aortic dissection and sudden death occurred 1 week after Ang II challenge. The median time window from the last MRI scan to sudden death was 2–3 days. The BAPN–Ang II challenge increased AD-associated mortality and thoracic aortic ruptures in Lum−/− mice compared with WT mice, but the aneurysm change of the two groups did not vary significantly. Therefore, we speculate that aortic pathogenesis due to the lack of LUM might only occur in thoracic aorta and might be directly involved in the events related to AD-associated mortality and aortic rupture without obviously influencing AA morphology. However, we agree with your opinion that the possibility of a missed window of aneurysm progression should be considered. 

8. Figure 6. How long was the treatment in experimental group? Are mice in Fig 4 and 6 from the same experimental treatment group? Please show representative WB images. IHC assay should be informative.

Reply: For the induction of AD in the experimental group, 3-week-old male mice were fed a regular diet and administered BAPN dissolved in drinking water (at 1 g/kg) per day for 4 weeks. At 7 weeks of age, the study group received micro-osmotic pump implants filled with Ang II at 1 μg/kg/min. 

Mice in Figures 4 and 6 were from the same experimental treatment group. 

In accordance with your suggestion, we have revised our Figure 6 to provide representative WB images and revised our Figure 4 to elaborate on the IHC staining. 

9. Line 236 in Discussion: “To the best of our knowledge, this is the first study to demonstrate LUM expression patterns in patients with AAD“. Please discuss PMID 22228989, 26998013, 32698686.

Reply: We have revised the indicated sentence as follows: “Our study results revealed LUM expression patterns in the intima and media of the ascending aorta of patients with AAD” and added the relevant discussion. 

Page 12 Line 274: Results of current study indicate that LUM expression patterns are in the intima and media of the ascending aorta of patients with AAD. Gu et al. identified LUM as a potential serum marker by using quantitative proteomics [8]. They further observed that LUM is expressed in the aortic medial layer in both healthy donors and patients with AAD [9]. However, LUM expression in the intimal layer was not reported. Another study using a mouse model to examine transcriptomic changes during aortic aneurysm development in Marfan syndrome identified differential expression patterns of LUM in SMC [13]. Other studies on LUM expression have focused on atherosclerosis and arterial calcification.

10. Fig 6: Was the expression of Lum increased in the aorta wall and the serum from the experimental WT group? Was Lum expression level associated with the risk/severity of AD in the WT experimental group?

Reply: We performed an additional analysis of Lum expression in the aortic wall of the experimental WT group; this group had significantly higher Lum expression than did the control WT mice.

Page 12 line 243: We also assessed Lumican protein expression patterns in WT mice with or without BAPN-Ang II challenges (Fig. 7A) and found that Lumican expression was significantly higher (2.1-fold, p < 0.05) in the BAPN-Ang II–challenged WT mice (Fig 7B).

Because most mice with AD in the experimental group had a sudden death, we could not obtain enough samples to explore the association between lumican levels and AD severity. However, our previous study, entitled “Level of Serum Soluble Lumican and Riks of Perioperative Complications in Patients Receiving Aorta Surgery” published in Plos One in March 2021, demonstrated that the serum soluble lumican levels can be a potential prognostic factor for predicting poor outcomes after aortic surgery in patients with AD. 

 

Reviewer #2: The manuscript entitled "Expression and Role of Lumican in Acute Aortic Dissection: A Human and Mouse Study" by Chen et al examined LUM expression 40 patterns in patients with AD and explored the molecular functions of LUM in a mouse AD model. The authors concluded that LUM expression is altered in patients with AAD, and Lum−/− mice exhibited augmented AD pathogenesis and LUM may be a novel target for AD therapy.

While the concept of the paper is of high importance towards the possibility of Lum as novel therapeutic target for AAA, There are major concerns with the manuscript:

1. While differences in Lum expression was observed between acute AD (AAD) and chronic AD (CAD), it is not clear how these tissue samples was characterized. Also, in Fig. 1, the staining is not clear and resolution is poor. In fact, in Fig. 1F, it seems that expression of Lum is increased in the adventitial layer.

Reply: We have accordingly revised Figure 1 to clarify our findings. The ascending aorta of patients with AAD was obtained to examine LUM expression through immunohistochemical staining (Fig. 1). LUM expression was particularly high in the aortic intima and media, but no significant expression was noted in the adventitial layer. 

Line 464: Figure 1 legend: Lum expression patterns in the ascending aorta of patients with acute aortic dissection (AAD). (A–C) Representative images of human ascending aorta stained with Masson’s trichrome, revealing patterns of collagen and elastic fibers. A, M, and E represent the adventitia, media, and endothelium layers, respectively. (D–F) Corresponding sections adjacent to the A–C sections and immunohistochemically stained with the human LUM antibody (R&D, AF2846). Black arrows indicate presence of LUM in the aortic intima and media. 

2. Figs. 4C-F in the present form is not adding much to the data. The authors need to add high magnification images and interpretation of the results.

Reply: We have revised our Figure 4 and interpretation of the results per your suggestion.

3. The authors reported higher TAD in the Lum KO mice, how does it correlate to the human AADs is not clear.

Reply: In this study, we demonstrated that LUM was expressed in intimal and medial layers of the ascending aorta of patients with AAD. Patients with AAD had higher serum s-LUM levels than that of patients with CAD. The AD mouse model further demonstrated that lack of Lum contributed to increased risks of AD-related mortality and thoracic aortic rupture and altered Tgf-β/Smad signaling and Mmps expression, but the aneurysm change of the two groups did not vary significantly. Therefore, our observations are consistent to the notion that aortic pathogenesis due to the lack of Lum might only occur in thoracic aorta and might be directly involved in the events related to AD-associated mortality and aortic rupture without obviously influencing AA morphology. We also found that Lum expression was significantly higher in the BAPN-Ang II–challenged WT than naive WT mice. Taken together, our findings suggest that Lumican is important to maintain the aortic structure to prevent aortic dissection and may be crucial in alleviating AD pathogenesis in that Lum-/- mice display severe pathogenesis upon induction of AAD by BAPN-Ang II challenge, which weakens cross-link of collagen fibrils in ECM. We speculate that LUM increase to response the aortic injury as a healing compensation both in human and mice. Therefore, the Increase of s-LUM is an important clinical biomarker of AD. However, further studies for explaining the regulation mechanism of LUM and clarifying its role in the human aorta are warranted.

We have revised our Discussion section in page 12 line 254.

Furthermore, our previous study, entitled “Level of Serum Soluble Lumican and Risks of Perioperative Complications in Patients Receiving Aorta Surgery” published in Plos One in March 2021, demonstrated that serum soluble lumican levels can be a potential prognostic factor for predicting poor outcomes after aortic surgery. 

4. It seems that most of the mice (18/22) died of TAD. It is not clear how the remaining mice were used.

Reply: The remaining mice were sacrifice after a 28-day Ang II challenge. The aortic tissue from the ascending aorta to the iliac bifurcation was resected for WB analysis (Page 12 Line 242).

5. It is not clear which part of the tissue (aorta?) and how many were used for Fig 6A. It is also not clear whether the authors used abdominal or the whole aorta.

Reply: Aortic tissue from the ascending aorta to the iliac bifurcation was resected and frozen in liquid nitrogen until further analysis (Page 12 Line 242). Moreover, we have revised our results to clarify our findings.

---

## [Decision Letter · Decision Letter 1]

28 Jun 2021

PONE-D-21-03961R1

Expression and Role of Lumican in Acute Aortic Dissection: A Human and Mouse Study

PLOS ONE

Dear Dr. Chu,

Thank you for submitting your manuscript to PLOS ONE. After careful consideration, we feel that it has merit but does not fully meet PLOS ONE’s publication criteria as it currently stands. Therefore, we invite you to submit a revised version of the manuscript that addresses the points raised during the review process.

We look forward to receiving your revised manuscript.

Kind regards,

Helena Kuivaniemi, MD, PhD

Academic Editor

PLOS ONE

Journal Requirements:

Additional Editor Comments (if provided):

Reviewers' comments:

Reviewer's Responses to Questions

**Comments to the Author**

1. If the authors have adequately addressed your comments raised in a previous round of review and you feel that this manuscript is now acceptable for publication, you may indicate that here to bypass the “Comments to the Author” section, enter your conflict of interest statement in the “Confidential to Editor” section, and submit your "Accept" recommendation.

Reviewer #1: All comments have been addressed

Reviewer #2: All comments have been addressed

2. Is the manuscript technically sound, and do the data support the conclusions?

Reviewer #1: Yes

Reviewer #2: Yes

3. Has the statistical analysis been performed appropriately and rigorously? 

Reviewer #1: Yes

Reviewer #2: N/A

4. Have the authors made all data underlying the findings in their manuscript fully available?

Reviewer #1: Yes

Reviewer #2: Yes

5. Is the manuscript presented in an intelligible fashion and written in standard English?

Reviewer #1: Yes

Reviewer #2: Yes

6. Review Comments to the Author

Reviewer #1: The authors have been responsive to the critiques. The revised manuscript is much improved. Here are minor comments:

o Fig 1 and Fig 4. In figure legends, please indicate the size of the bar. The size in the figures are too small to read.

o It is interesting that the expression of Lum is increased in the aorta. Since AD occurs preferentially at the ascending aorta, do you see differential expression of Lum in thoracic and abdominal aorta?

Reviewer #2: The authors have addressed all the concerns satisfactorily. The Reviewer does not have additional concerns.

7. PLOS authors have the option to publish the peer review history of their article (what does this mean?). If published, this will include your full peer review and any attached files.

Reviewer #1: No

Reviewer #2: **Yes: **Chetan P Hans

---

## [Author Response · Author response to Decision Letter 1]

7 Jul 2021

July 1, 2021

Editor-in-Chief

PLOS ONE

PONE-D-21-03961 Expression and Role of Lumican in Acute Aortic Dissection: A Human and Mouse Study

Dear Editor:

On behalf of the authors of this manuscript, we thank you, associate editor and 2 reviewers very much for your thorough review of this manuscript. Your suggestions were very thoughtful and instructive. The reviewers identified some serious and important concerns that we have not considered while making this study. We revised the manuscript according to the comments of the reviewers’ point-by-point as follows. We hope that the revised manuscript will retain your attention and you will judge the revised manuscript to be suitable for publication in “PLOS ONE”. Please forward the revised manuscript to the reviewers and reconsider the work for publication. 

Yours sincerely,

First authors: Shao-Wei Chen 

Corresponding author: Pao-Hsien Chu

Reviewer #1: The authors have been responsive to the critiques. The revised manuscript is much improved. Here are minor comments:

Fig 1 and Fig 4. In figure legends, please indicate the size of the bar. The size in the figures are too small to read.

Reply: We have revised the size bar in Fig 1 and Fig 4, and we also indicated the size of the bar in the figure legend.

It is interesting that the expression of Lum is increased in the aorta. Since AD occurs preferentially at the ascending aorta, do you see differential expression of Lum in thoracic and abdominal aorta?

Reply: Thank you for your valuable comment. Unfortunately, because of the limited number of WT mice with AD in the experimental group, we could not obtain enough samples to explore the different expressions of Lum in the thoracic and abdominal aorta. We will further explore this interesting issue in our future study. Thanks again. 

Reviewer #2: The authors have addressed all the concerns satisfactorily. The Reviewer does not have additional concerns.

Reply: Thank you for your review.

---

## [Editor Report · Decision Letter 2]

13 Jul 2021

Expression and Role of Lumican in Acute Aortic Dissection: A Human and Mouse Study

PONE-D-21-03961R2

Dear Dr. Chu,

We’re pleased to inform you that your manuscript has been judged scientifically suitable for publication and will be formally accepted for publication once it meets all outstanding technical requirements.

Congratulations!

Kind regards,

Helena Kuivaniemi, MD, PhD

Academic Editor

PLOS ONE
---

## [Editor Report · Acceptance letter]

16 Jul 2021

PONE-D-21-03961R2 

Expression and Role of Lumican in Acute Aortic Dissection: A Human and Mouse Study 

Dear Dr. Chu:

I'm pleased to inform you that your manuscript has been deemed suitable for publication in PLOS ONE. Congratulations! Your manuscript is now with our production department. 

Kind regards, 

on behalf of

Professor Helena Kuivaniemi 

Academic Editor

PLOS ONE